# Cell cycle-dependent cues regulate temporal patterning of the *Drosophila* central brain neural stem cells

Gonzalo N Morales Chaya[1,2], Mubarak Hussain Syed[1]*

[1]Neural Diversity Lab, Department of Biology, University of New Mexico, Albuquerque, United States; [2]Institute of Neuroscience, Howard Hughes Medical Institute, University of Oregon, Chevy Chase, United States

## eLife Assessment

This manuscript reports **important** findings indicating that cell cycle progression and cytokinesis both contribute to the transition from early to late neural stem cell fates. Loss-of-function experimental evidence **convincingly** shows that disrupting the cell cycle or cytokinesis can alter cell fate. This work sets the stage for future investigations into the underlying mechanisms linking the cell cycle to the expression of temporal factors controlling cell fate.

*For correspondence:
flyguy@unm.edu

**Competing interest:** The authors declare that no competing interests exist.

**Abstract** During nervous system development, diverse types of neurons and glia are sequentially generated by self-renewing neural stem cells (NSCs). Temporal changes in gene expression within NSCs are thought to regulate neural diversity; however, the mechanisms regulating the timing of these temporal gene transitions remain poorly understood. *Drosophila* type 2 NSCs, like human outer radial glia, divide to self-renew and generate intermediate neural progenitors, amplifying and diversifying the population of neurons innervating the central complex, a brain region crucial for sensorimotor coordination. Type 2 NSCs express over a dozen genes temporally, broadly classified as early- and late-expressed genes. A conserved gene, *seven-up*, mediates early-to-late gene expression by activating ecdysone receptor (EcR) expression. However, the timing of EcR expression and, consequently, the transition from early-to-late gene expression remain unknown. This study investigates whether intrinsic mechanisms of cell cycle progression and cytokinesis are required to induce the NSC early-late transition. By generating mutant clones that arrest the NSC cell cycle or block cytokinesis, we show that both processes are necessary for the early-to-late transition. When NSCs are cell cycle or cytokinesis arrested, the early gene Imp fails to be downregulated and persists in the old NSCs, while the late factors EcR and Syncrip fail to be expressed. Furthermore, we demonstrate that the early factor Seven-up is insufficient to drive the transition, despite its normal expression in cell cycle- or cytokinesis-inhibited NSCs. These results suggest that both cell-intrinsic (cell cycle/cytokinesis) and -extrinsic (hormone) cues are required for the early-late NSC gene expression transition.

## Introduction

From insects to humans, multipotent neural stem cells (NSCs) generate diverse neurons and glia, which mediate proper nervous system function. Both spatial and temporal patterning programs regulate the diversity of neurons generated during nervous system development (*Doe, 2017*; *El-Danaf et al., 2023*; *Enriquez et al., 2015*; *Erclik et al., 2017*; *Hamid et al., 2023*; *Kohwi and Doe, 2013*; *Malin and Desplan, 2021*; *Oberst et al., 2019*; *Sen, 2023*; *Skeath and Thor, 2003*; *Wani et al., 2023*).

While much is known about the regulation of spatial identity, the mechanisms underlying temporal identity are not entirely understood. Elucidating the processes through which NSCs undergo temporal patterning, enabling them to produce diverse neurons and glial cells in the nervous system, is critical for understanding brain development and disease. Studies on the *Drosophila* nervous system have identified temporal identity genes that express temporally in NSCs and diversify neural cell types (*Bayraktar and Doe, 2013*; *Eldred et al., 2018*; *Isshiki et al., 2001*; *Li et al., 2013*; *Maurange et al., 2008*; *Syed et al., 2017*; *Yang et al., 2016*; *Chaya et al., 2025*). Similar temporal identity programs were later shown to regulate vertebrate retinal and cortical cell-type specification (*Alsiö et al., 2013*; *Elliott et al., 2008*; *Javed et al., 2023*; *Kohwi and Doe, 2013*; *Koo et al., 2023*; *Pebworth et al., 2021*; *Telley et al., 2019*). In *Drosophila* embryonic NSCs, these gene transitions were thought to be intrinsically regulated; however, our recent findings in larval central brain NSCs identified the role of extrinsic steroid growth hormone ecdysone in the early-to-late gene transition (*Syed et al., 2017*). Recently, thyroid hormone signaling has been identified as regulating the early-to-late born cone cell type in human retinal organoid cultures (*Eldred et al., 2018*). Over the past few years, various temporally expressed genes have been identified in both invertebrates and vertebrates; however, the factors that determine the precise timing of these gene transitions remain poorly understood.

Based on division patterns, the *Drosophila* NSCs can be classified as type 0, type 1, or type 2 NSCs. Type 0 NSCs generate a single post-mitotic neuron with each division and have only been observed in embryonic stages (*Arefin et al., 2020*; *Baumgardt et al., 2014*). Type 1 NSCs generate a ganglion mother cell (GMC) with each division; each GMC makes two post-mitotic neurons (*Doe, 2017*; *Doe and Goodman, 1985*; *Rossi and Desplan, 2017*). While type 1 NSCs comprise the majority of NSCs in the brain, type 2 NSCs—though limited to just 16—generate more complex and diverse progeny by exhibiting a more complex division pattern, producing approximately 5000 neurons of at least about 160 distinct cell types (*Epiney et al., 2025*). Type 2 NSCs divide to self-renew and generate intermediate neural progenitors (INPs); each INP undergoes a series of four to six self-renewing divisions to produce four to six GMCs, which each generate two neurons (*Bello et al., 2008*; *Boone and Doe, 2008*; *Bowman et al., 2008*; *Wang et al., 2014*). Recent temporal RNA-sequencing experiments have revealed that type 2 NSCs undergo a gene expression cascade involving transcription factors (TFs) and RNA-binding proteins (RBPs). In early larvae 0–60 hr after larval hatching (ALH), the NSCs express TFs, Seven-up (Svp), Castor (Cas), Chinmo, and RBPs, IGF-II mRNA-binding protein (Imp), and Lin-28. In late larvae (60–120 hr ALH), the NSCs express TFs, Ecdysone receptor (EcR), Ecdysone-induced protein 93F (E93), Broad and RBP, Syncrip (Syp) (*Ren et al., 2017*; *Syed et al., 2017*). The early-expressed conserved TF Svp regulates the expression of EcR, thereby rendering NSCs competent to respond to the extrinsic hormonal signal, which triggers the early-late gene expression switch (*Syed et al., 2017*). While the role of the extrinsic factor Ecdysone in triggering the early-late transition is understood, nothing is known about how intrinsic factors drive this transition and what regulates the timing of these events.

In vertebrate cortical development, the cell cycle plays a key role in regulating a neural progenitor's ability to respond to extrinsic cues. Transplanted progenitors in a late-stage host exhibit different responses to these cues depending on their cell cycle stage at the time of transplantation (*McConnell and Kaznowski, 1991*; *Ohnuma and Harris, 2003*). In *Drosophila*, several studies have demonstrated a relationship between the cell cycle and temporal gene expression programs in NSCs. Interestingly, this relationship differs markedly between embryonic and larval stages. In embryonic NSCs, the classic temporal cascade—Hunchback>Kruppel>Pdm1/2>Castor—is largely cell-intrinsic and can proceed in isolated, G2-arrested cells (*Grosskortenhaus et al., 2005*). While the transition from Hunchback to Kruppel requires cytokinesis, subsequent transitions (Kruppel to Pdm1/2 to Castor) occur even when cell cycle progression and cytokinesis are blocked.

In contrast, larval NSCs exhibit temporal transitions from early factors (Imp/Chinmo) to late factors (Syp/Broad) that depend on the cell's metabolic state and progression through the G1/S phase of the cell cycle (*van den Ameele and Brand, 2019*). Notably, at the end of larval neurogenesis, late temporal programs schedule the end of neurogenesis, and NSCs undergo cell cycle exit or apoptosis. These late-stage transitions are governed by the Svp-mediated induction of the EcR and its downstream effector E93 (*Pahl et al., 2019*; *Homem et al., 2014*; *Maurange et al., 2008*; *Syed et al., 2017*). However, whether Svp and EcR expression—and thus the timing of early-to-late transitions—are dependent on cell cycle progression remains unknown. Furthermore, it is not yet clear whether

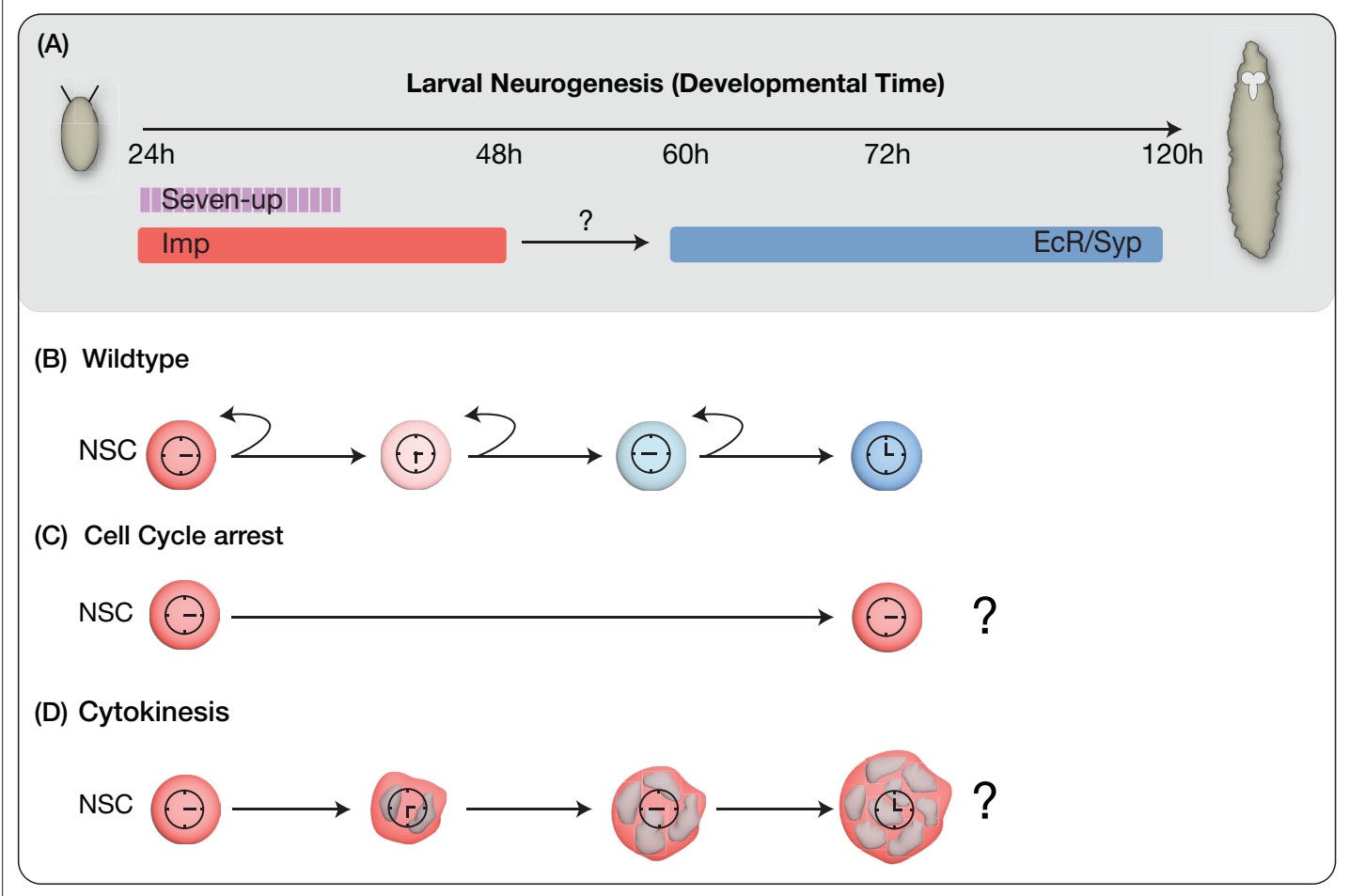

**Figure 1.** Models for early-to-late temporal progression transition of type 2 neural stem cells (NSCs). (**A**) Depicts timing of NSC temporal factor expression in type 2 NSCs during *Drosophila* larval development. (**B**) Wild-type NSCs undergo normal early-to-late temporal factor progression. (**C**) Cell cycle progression as a regulator to allow timely expression of factors in NSCs. (**D**) Cytokinesis-dependent regulation of temporal gene expression.

similar temporal and cell cycle interactions occur in type 1 and type 2 NSCs. Moreover, it is unclear whether the timing of NSC gene expression is driven by organismal cues (e.g. hormones or signaling pathways), internal cues (e.g. a cell cycle counting clock), or both. Here, we utilize larval central brain NSCs to investigate whether cell cycle and cytokinesis regulate the timing of the temporal gene expression program.

## Results

### Cell cycle and cytokinesis are both required for the early-late temporal factor switch in type 2 NSCs

Our previous work revealed that in type 2 NSCs, the transition from early-to-late temporal factors occurs around 55 hr ALH (*Syed et al., 2017*). To better understand the mechanisms regulating the precise timing of these gene expression shifts, we next focused on the roles of the cell cycle and cytokinesis. While the cell cycle has been shown to influence temporal patterning in some systems, it appears to be dispensable in others, suggesting context-dependent roles in regulating these transitions (*van den Ameele and Brand, 2019*; *Grosskortenhaus et al., 2005*; *Syed et al., 2017*; *Ray and Li, 2022*). To investigate whether the temporal progression of type 2 NSCs requires cell cycle progression, we blocked the NSC cell cycle or cytokinesis in all type 2 NSCs using the driver Wor-GAL4, Ase-GAL80, starting at 0 hr ALH (the experimental model is shown in *Figure 1*). We inhibited the cell cycle in all type 2 NSCs by knocking down Cyclin-dependent kinase 1 (Cdk1), a cdc2 kinase

that forms heterodimers with Cyclin A and Cyclin B and is required for the execution of mitosis. Larvae were collected at 0 hr ALH, and temporal factor expression was assayed at 72 hr ALH. The *Cdk1RNAi* effectively blocked the cell cycle, as we could not observe any GFP-labeled progeny near the NSCs, and the NSC cell volume was larger than that of the control, which has also been attributed to normal cell cycle progression (*Hartenstein et al., 1987*; *Figure 2—figure supplement 1*). While all control type 2 NSC lineages showed normal temporal factor expression progression, *Cdk1RNAi* type 2 NSCs exhibited prolonged expression of the early factor Imp into later stages of development and failed to upregulate the late factors Syp and EcR (*Figure 2—figure supplement 2*), confirming that cell cycle is essential for the early-to-late gene transition.

To further ensure the cell specificity of this phenotype, better quantify, and allow NSCs to develop in an otherwise wild-type brain, we used the flip-out approach that induced random single-cell clones of type 2 NSCs. We induced random clones by giving a heat shock at 0 hr ALH using the genotype *hsflp, 10xUAS-mCD8GFP; Act >FRT-stop-FRT>GAL4*. Both control RNAi (*UAS-mCherryRNAi*) and experimental (*UAS-Cdk1RNAi*) clones were labeled by GFP, and type 2 hits were confirmed by lack of the type 1 Asense marker (*Figure 2—figure supplement 1E–G*). After heat shock, we assayed the clones in 72 hr ALH larvae. In the control type 2 NSCs, as anticipated at 72 hr ALH, minimal or no expression of the early factor Imp was observed, alongside high expression of late factors EcR and Syp (*Figure 2A–C*; quantified in *Figure 2J*). In contrast, the experimental type 2 NSCs, where the cell cycle was blocked, exhibited persistent expression of the early factor Imp and no expression of late factors EcR and Syp at 72 hr ALH (*Figure 2D–F*; quantified in *Figure 2J*), thus confirming that cell cycle progression is indispensable for the early-late switch in temporal factor gene expression.

The failure to switch from early to late factors could arise from blocking the cell cycle progression (as part of a gene expression 'clock' or 'timer') or from preventing cytokinesis, resulting in a daughter cell that may deliver feedback signals or partition transcriptional regulators out of the NSC. To discern between these possibilities, we inhibited cytokinesis, halting NSC progeny production while leaving its cell cycle unaffected by expressing pavarotti RNAi (*UAS-pavRNAi*), a well-established cytokinesis inhibitor whose knockdown produces large, multinucleated cells (*Grosskortenhaus et al., 2005*; *Adams et al., 1998*). Similar to our previous observations (*Figure 2—figure supplement 2*), control type 2 NSCs showed no Imp expression and normal EcR/Syp expression (*Figure 2A–C*; quantified in *Figure 2J*), while the experimental *pavRNAi* type 2 NSCs resulted in large multinucleated NSCs (*Figure 2—figure supplement 1C–C'*) that maintained Imp expression and lacked EcR/Syp expression (*Figure 2G–I*; quantified in *Figure 2J*). These results were consistent with blocking cytokinesis simultaneously in all type 2 NSCs (*Figure 2—figure supplement 2*). We conclude from these experiments that both cell cycle progression and cytokinesis are essential for the early-late gene expression switch in type 2 NSCs (*Figure 2K*).

## The switching factor Svp is normally expressed in cell cycle-arrested NSCs

A conserved nuclear hormone receptor, Svp, mediates the early-to-late gene switch by regulating EcR expression (*Dillon et al., 2024*; *Maurange et al., 2008*; *Syed et al., 2017*). The *svp* mutant type 2 NSCs stay in their early gene expression module and fail to exit the cell cycle at the end of larval life. The mutant type 2 NSCs fail to downregulate the expression of early factors Chinmo and Imp and show no expression of the late factors EcR, Broad, Syncrip, and E93. While Svp shows its peak expression at 18–24 hr ALH, EcR expression starts at around 55 hr ALH (*Dillon et al., 2024*; *Ren et al., 2017*; *Syed et al., 2017*). Although *svp* is required for EcR expression, what regulates the precise timing of EcR expression around 55 hr ALH remains unknown. The lack of EcR expression in cell cycle-arrested type 2 NSCs could be due to either defective Svp expression or the lack of a cell cycle counter or both.

To test whether cell cycle-arrested NSCs fail to express Svp and thus produce a phenotype similar to the svp mutant, we assayed Svp expression in type 2 cell cycle-arrested NSCs. Since Svp expression is transient in these NSCs, we used a *svpLacZ* line, similar to our previous work (*Syed et al., 2017*), to achieve more stable protein expression. Upon counting LacZ-positive type 2 NSCs at 48 hr ALH, nearly all control type 2 NSCs showed *svpLacZ* expression (*Figure 3A*), and cell cycle-arrested animals displayed similar expression levels with no statistically significant difference from controls (*Figure 3B*, quantified in *Figure 3C*). We conclude that the timing of Svp expression is normal in cell cycle-arrested NSCs.

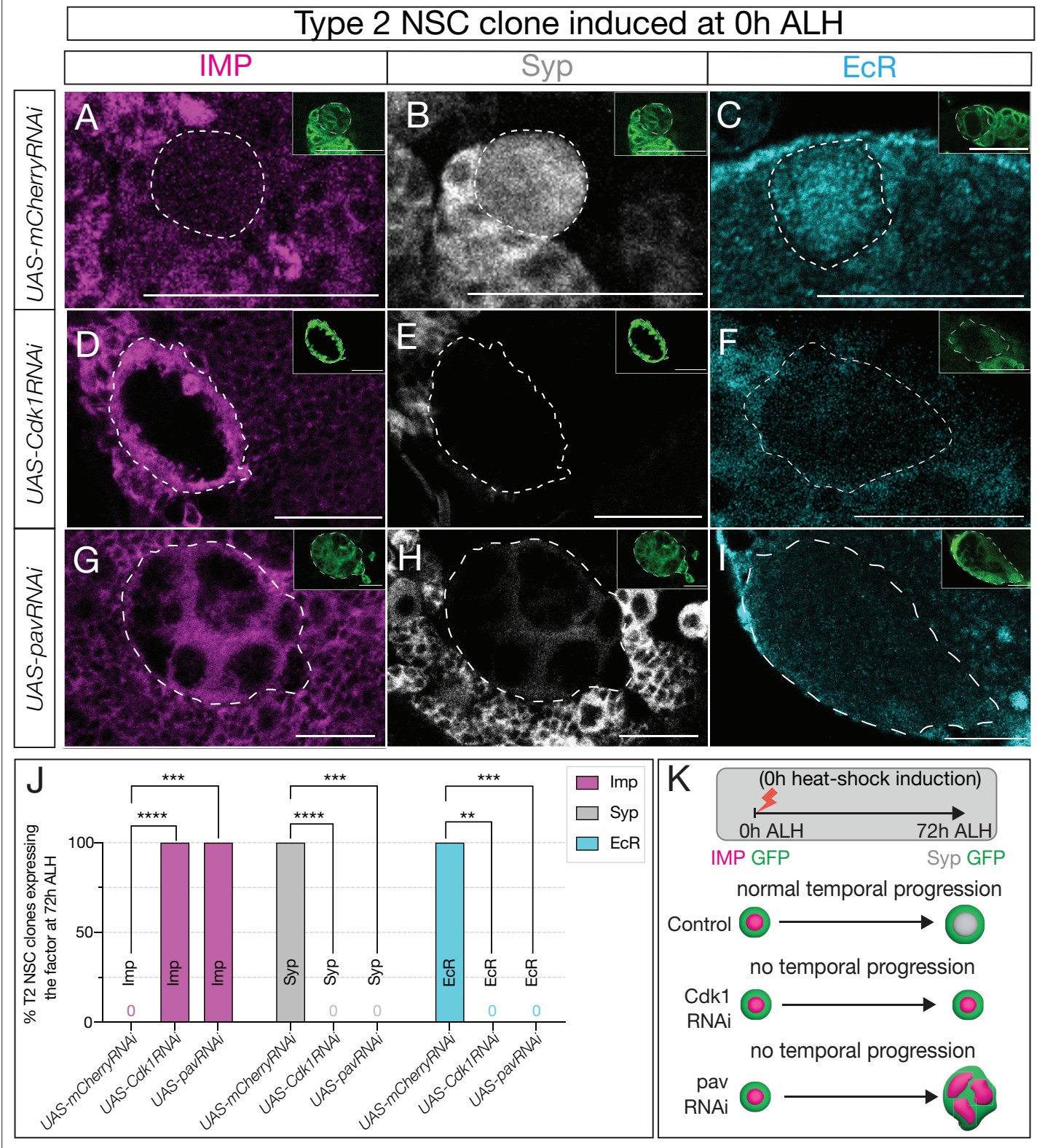

**Figure 2.** Cell cycle and cytokinesis are required for temporal gene expression progression in type 2 neural stem cells (NSCs). (**A–C**) Type 2 NSC control *mCherryRNAi* clones (circled) at 72 hr after larval hatching (ALH) show normal temporal gene expression progression. Early factor Imp is off, and late factors Syp and EcR are on at 72 hr ALH (**D–F**) *Cdk1RNAi* type 2 NSC clone fails to downregulate early factor Imp, and late factors Syp and EcR are not turned on. (**G–I**) Similarly, cytokinesis-blocked type 2 NSC clones using *pavRNAi* fail to downregulate Imp and upregulate Syp and EcR. (**J**) Quantification of early (Imp) and late (EcR, Syp) temporal factor expression in *mCherryRNAi*, *Cdk1RNAi*, and *pavRNAi* type 2 NSCs (n≥4 clones per genotype).

*Figure 2 continued on next page*

*Figure 2 continued*

Statistical analysis by Fisher's exact test: Imp: 0/5 vs 11/11 clones in *Cdk1RNAi* (p=0.0002), 0/5 vs 8/8 in *pavRNAi* (p=0.0008); EcR: 6/6 vs 0/4 in *Cdk1RNAi* (p=0.0048), 6/6 vs 0/7 in *pavRNAi* (p=0.0006); Syp: 5/5 vs 0/11 in *Cdk1RNAi* (p=0.0002), 5/5 vs 0/8 in *pavRNAi* (p=0.0008). Only one clone was found per animal. (**K**) Representation of experimental layout for inducing clones at 0 hr ALH and analysis at 72 hr ALH. Type 2 NSCs are identified as Asense-negative large cells expressing UAS-mcd8::GFP (green insets). Scale bars represent 20 μm.

The online version of this article includes the following source data and figure supplement(s) for figure 2:

**Source data 1.** Number of T2 NSC clones expressing temporal factors on 0h ALH heat-shocked animals.

**Figure supplement 1.** *pav* and *Cdk1RNAi* type 2 neural stem cell (NSC) clones are larger in volume.

**Figure supplement 1—source data 1.** Raw clone volume values per genotype.

**Figure supplement 2.** Cell cycle and cytokinesis inhibit the early-to-late transition of temporal factors in type 2 neural stem cells (NSCs).

## Svp and cell cycle progression are independently required for the early-late temporal factor switch

Since Svp expression was normal in cell cycle-arrested type 2 NSCs, we wondered whether the cell cycle acts as a timer for the precise EcR expression following Svp expression. To address this question, we generated type 2 NSC clones expressing either *Cdk1RNAi* or *pavRNAi* at 42 hr ALH (see Materials and methods) well after the Svp expression window (18–24 hr ALH). We assayed for expression of NSC temporal markers at 72 hr ALH. If the transition fails to occur, it will confirm that Svp primes EcR expression, but the intrinsic cell cycle clock determines the precise timing. Conversely, if the transition occurs normally, cell cycle/cytokinesis plays a role in early gene expression (before 42 hr), but not in later, post-Svp expression. Compared to control type 2 NSC clones that showed normal Imp down-regulation and EcR/Syp expression at 72 hr ALH (*Figure 4A–C*; quantified in *Figure 4J*), Cdk1 and pav mutant type 2 NSC clones had persistent Imp expression and no EcR/Syp expression (*Figure 4D and I*; quantified in *Figure 4J*), confirming that cell cycle progression post-Svp expression is critical for precise timing of EcR and early-to-late gene transition (*Figure 3K*).

## Cell cycle and cytokinesis are required for the early-late temporal factor switch in type 1 NSCs

So far, we have focused on type 2 NSCs, which are relatively few; however, the majority of larval NSCs are type 1 NSCs, which express the same early and late temporal factors as type 2 NSCs (*Syed*

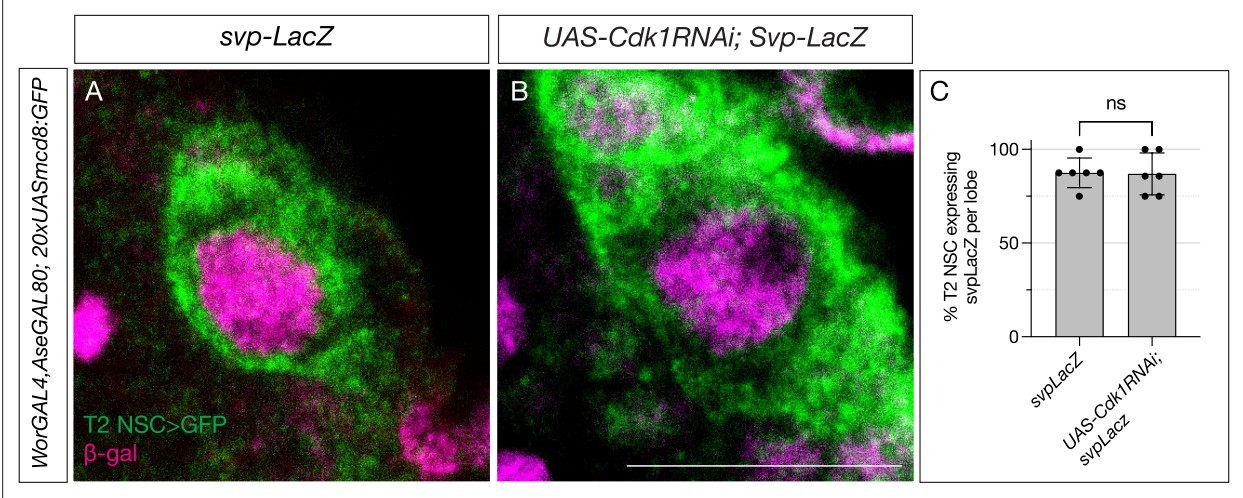

**Figure 3.** Switching factor Svp is expressed normally in cell cycle-arrested type 2 neural stem cells (NSCs). (**A**) Control type 2 NSCs marked in green show *Svp-LacZ* expression at 48 hr after larval hatching (ALH). (**B**) Cell cycle-arrested (*Cdk1RNAi*) type 2 NSCs express *Svp-LacZ* similar to the control. (**C**) Quantification of LacZ expression in control and cell cycle-blocked type 2 NSCs, n=6 for each genotype. Unpaired t-test, p-value = 0.9168. Scale bars represent 10 μm.

The online version of this article includes the following source data for figure 3:

**Source data 1.** Raw values showing percent of T2 NSC expressing svp-LacZ per brain.

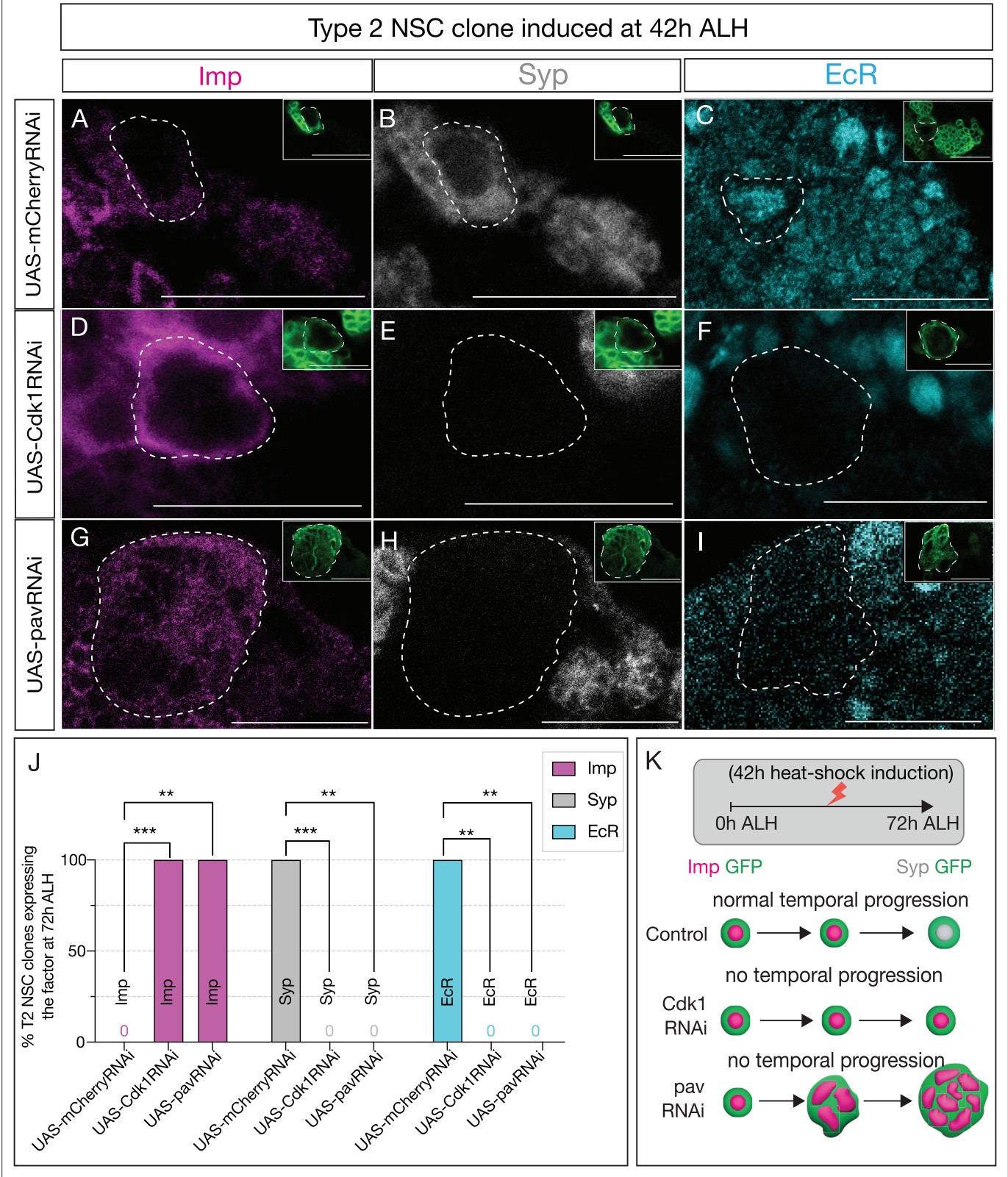

**Figure 4.** Early Svp expression is not sufficient to drive temporal factor progression in type 2 neural stem cells (NSCs). (**A–C**) Type 2 NSC control *mCherryRNAi* clones (circled) induced at 42 hr after larval hatching (ALH) and stained at 72 hr ALH show normal progression of temporal factors. Early factor Imp is off, and late factors Syp and EcR are on. (**D–F**) *Cdk1RNAi* clones show consistent failure to downregulate early factor Imp and activate Syp and EcR. (**G–I**) Likewise, *pavRNAi* clones fail to express EcR and Syp and consistently express early factor Imp. (**J**) Quantification of early

*Figure 4 continued on next page*

*Figure 4 continued*

(Imp) and late (EcR, Syp) temporal factor expression in control (*UAS-mCherryRNAi*), *UAS-Cdk1RNAi*, and *UAS-pavRNAi* type 2 NSCs (n≥4 clones per genotype). Statistical analysis by Fisher's exact test: Imp: *Cdk1RNAi* vs control p=0.0002, *pavRNAi* vs control p=0.0079; EcR: *Cdk1RNAi* vs control p=0.0079, *pavRNAi* vs control p=0.0079; Syp: *Cdk1RNAi* vs control p=0.0002, *pavRNAi* vs control p=0.0079. Only one clone was found per animal. (**K**) Representation of experimental layout. Clones are identified as Asense-negative large cells expressing UAS-mcd8::GFP (green insets). Scale bars represent 20 µm.

The online version of this article includes the following source data for figure 4:

**Source data 1.** Number of T2 NSC clones expressing temporal factors on 42h ALH heat-shocked animals.

et al., 2017). The timing of EcR expression and the transition from early-to-late gene expression also matches the type 2 NSC pattern (*Syed et al., 2017*). Thus, we sought to determine if cell cycle and cytokinesis also regulate temporal gene transitions in type 1 NSCs.

Similar to our previous experiments, we generated clones using the genotype hsFLP, UAS-mCD8GFP; Act>FRT-stop-FRT>GAL4/*UAS-Cdk1RNAi* or *UAS-mCherryRNAi* and assayed GFP and Asense-positive type 1 NSCs (*Figure 2—figure supplement 1E–G'*). Control RNAi induced at 0 hr and assayed at 72 hr showed normal downregulation of Imp and expression of the late factors Syp (*Figure 5A–C and K–M*; quantified in *Figure 5J and T*). In contrast, *Cdk1* or *pavRNAi* type 1 NSC clones induced at 0 hr and assayed at 72 hr maintained Imp expression and lacked EcR/Syp expression (*Figure 5D–I*; quantified in *Figure 5J*).

Similarly, *Cdk1* or *pavRNAi* induced at 42 hr—after the Svp expression window—and assayed at 72 hr also maintained Imp expression and lacked EcR/Syp expression (*Figure 5N–S*; quantified in *Figure 5T*). This suggests that most central brain NSCs possess a cell cycle- and cytokinesis-dependent, cell-intrinsic mechanism that regulates the Imp/Syp expression transition. Our studies indicate that all central brain NSCs require cell cycle and cytokinesis, together with the switching factor Svp expression, to drive the early-late temporal factor expression transition.

## Discussion

Temporal patterning plays a crucial role in specifying the diverse cell types of the nervous system. While many temporally expressed genes have been identified in both *Drosophila* and vertebrate neural progenitors (*Javed et al., 2023*; *Koo et al., 2023*; *Pebworth et al., 2021*; *Syed et al., 2017*; *Telley et al., 2019*; *Walsh and Doe, 2017*), what governs the timing of these gene expression transitions is not clearly understood. Is there an intrinsic timer within NSCs that counts and regulates the precise timing of temporal gene transitions? Is such a mechanism linked to cell cycle progression and cytokinesis? How are NSC intrinsic and extrinsic environmental cues coupled? This study investigated the cell-intrinsic mechanisms regulating the early-to-late temporal gene expression transitions. Using *Drosophila* larval central brain NSCs, we identified that both cell cycle and cytokinesis-dependent cell-intrinsic timers regulate the progression of gene expression. In cytokinesis-blocked NSCs, the transition from early Imp expression to late EcR and Syp expression was impeded in both type 1 and type 2 NSCs. Upon temporal cell cycle block at 42 hr ALH, we showed that normal cell cycle progression is still essential for EcR expression despite the Svp expression being normal in the NSCs. These findings provide insights into the possible mechanisms regulating the temporal gene transition timing and also reveal the complex interplay of the cell cycle, cell-intrinsic genes, and cell-extrinsic hormonal cues in determining the precise timing of neuron-type production within neural progenitors.

The possible role of cell cycle and cytokinesis in regulating temporal patterning and neuronal identity has been studied in vertebrates and invertebrates (*Doe, 2008*; *Grosskortenhaus et al., 2005*; *Hagey et al., 2020*; *Kawaguchi, 2019*; *van den Ameele and Brand, 2019*). In vertebrates, the G2/M phase is required for the sequential generation of layer-specific cortical neurons. Knockdown of G2/M regulators has been shown to prevent radial glial cells from maintaining a 'young', multipotent state, leading to premature differentiation and a bias toward the production of early-born neurons (*Hagey et al., 2020*; *Kawaguchi, 2019*). In the case of *Drosophila* embryonic NSCs, which sequentially express Hunchback, Kruppel, Pdm1, and Castor, cell cycle is dispensable for these transitions. The initial Hunchback to Kruppel expression requires cytokinesis, while the Kruppel to Pdm1 to Castor transition occurs normally in G2-arrested NSCs (*Grosskortenhaus et al., 2005*). In contrast, our studies on larval NSCs show that both cell cycle and cytokinesis are essential for the early Imp/Chinmo to late

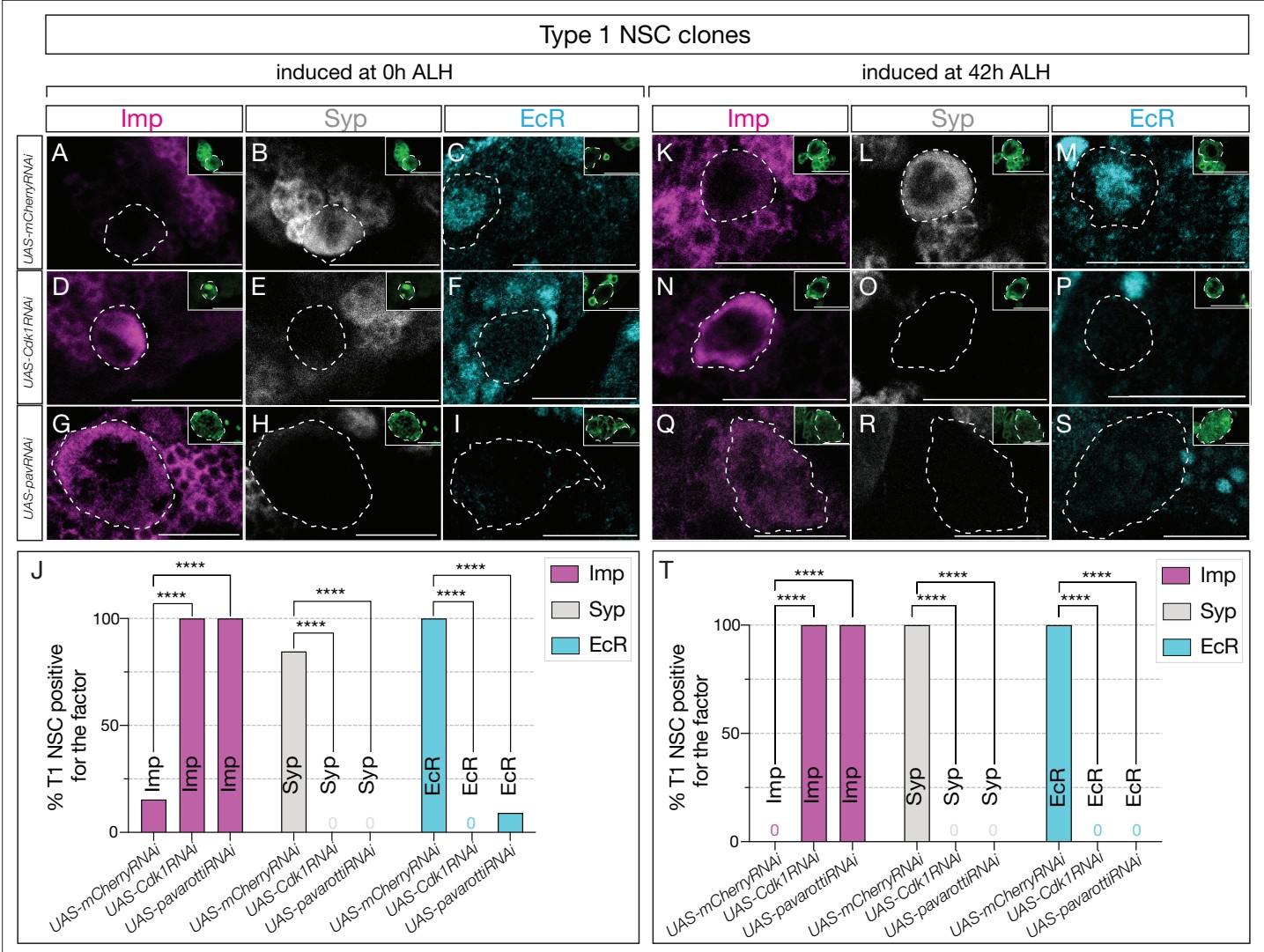

**Figure 5.** Cell cycle and cytokinesis are required for temporal progression in type 1 neural stem cells (NSCs). (**A–I**). Clones were induced at 0 hr after larval hatching (ALH), and for (**K–S**), clones were induced at 42 hr ALH. (**A–C**) Type 1 NSC control RNAi clones show normal expression of temporal factor progression. (**D–F**) Cell cycle-blocked type 1 NSC clones using Cdk1RNAi show failure to upregulate EcR and Syp, and consistent expression of early factor Imp. (**G–I**) *pavRNAi* type I NSC clones fail to downregulate early factor Imp, and late factors Syp and EcR fail to express. (**K–M**) Control type 1 NSC clones induced at 42 hr ALH show normal expression of early and late factors, while *Cdk1RNAi* clones (**N–P**) and *pavRNAi* clones (**Q–S**) fail to downregulate Imp and fail to activate late factors Syp and EcR. (**J, T**) Quantification of early and late factor expression of 0 hr and 42 hr ALH–induced clones. (**J**) In clones induced at 0 hr ALH, Imp was expressed in 10/10 *Cdk1RNAi* clones (2/13 control, p=0.000067) and 12/12 *pavRNAi* clones (2/13 control, p=0.000020). EcR was not detected in *Cdk1RNAi* (0/11) or *pavRNAi* clones (0/11), while it was expressed in 12/12 control clones (p=0.000001 for both). Syp was absent in *Cdk1RNAi* (0/10) and *pavRNAi* clones (0/12), while present in 11/13 control clones (p=0.000067 and p=0.000020, respectively). (**T**) In clones induced at 42 hr ALH, Imp expression persisted in 10/10 *Cdk1RNAi* (0/10 control, p=0.000011) and 13/13 *pavRNAi* clones (0/10 control, p=0.000001). EcR expression was not detected in *Cdk1RNAi* (0/8) or *pavRNAi* clones (0/13), compared to 10/10 control (p=0.000023 and p=0.000001). Syp expression was not detected in *Cdk1RNAi* (0/10) or *pavRNAi* clones (0/13) but was regular in 10/10 control (p=0.000011 and p=0.000001). Only one clone was found per animal. For each genotype, n≥8. Type 1 clones are identified as Asense-positive, large cells expressing mcd8::GFP (green insets). Scale bars represent 20 μm.

The online version of this article includes the following source data for figure 5:

**Source data 1.** Number of T1 NSC clones expressing temporal factors on 0h and 42h ALH heat-shocked animals.

EcR/Syncrip gene transition. The temporal gene progressions of embryonic NSCs occur normally in isolated NSCs, suggesting that these gene transitions are regulated intrinsically. In contrast, larval NSCs express EcR and require extrinsic growth hormone, ecdysone, for the transition from early-to-late gene expression. Our findings suggest that NSCs with more complex division patterns and

dividing in situations where the growth and division patterns need to be coordinated require both cell-intrinsic and -extrinsic cues. Studies on larval optic lobe NSCs thus far suggest that their temporal gene expression cascade is regulated cell-intrinsically, akin to embryonic NSCs (*Konstantinides et al., 2022*; *Li et al., 2013*). However, it remains unclear whether these temporal transitions depend on progression through the cell cycle/cytokinesis or both. This raises the intriguing question of whether optic lobe NSCs follow an embryonic-like model, in which temporal patterning occurs independently of cell division, or resemble larval central brain NSCs, where temporal progression is tightly coupled to the cell cycle and cytokinesis.

Through the cell cycle, neural progenitors generate neurons populating distinct layers in the verte-brate cortex (*Borrell and Calegari, 2014*; *Hagey et al., 2020*; *Ohtsuka and Kageyama, 2019*; *Salo-moni and Calegari, 2010*); however, the relationship between cell cycle and temporal patterning has remained elusive. Given that the cell cycle is essential for temporal gene transitions, it could be one potential way of timing the generation of distinct progeny with precise temporal control. While we were able to demonstrate the cell-intrinsic roles of cell cycle in timing temporal gene transitions, given the fact that NSC resides in a glial niche and in close proximity to its lineages (*Doe, 2008*), and the cell cycle-blocked lineages have few or no progeny, we cannot rule out the possibility that type 2 NSCs with cell cycle arrest failed to undergo normal temporal progression indirectly due to a lack of feedback signaling from their progeny. Additionally, we note that in some large multinu-cleated *pavRNAi* clones, a weak EcR signal can sometimes be observed near the cell membrane or within cytoplasmic regions (*Figure 2I*). However, the nuclear compartments of these polynucleated cells—where EcR is normally localized and active—show no detectable EcR staining. The origin of this membrane-proximal EcR signal remains unclear; it may reflect nonspecific antibody binding, altered EcR trafficking or degradation in cytokinesis-defective cells, or imaging artifacts arising from the large and irregular morphology of these clones. Importantly, the absence of nuclear EcR in the scored NSCs supports our conclusion that both cell cycle progression and cytokinesis are required for the acquisi-tion of nuclear EcR expression and for the early-to-late temporal transition.

We acknowledge that we cannot formally exclude gene-specific, cell cycle-independent roles of Cdk1 or Pav. However, several lines of evidence favor the interpretation that disruption of cell cycle progression or cytokinesis underlies the phenotype. First, Cdk1 and Pav are canonical regula-tors of mitosis and cytokinesis, respectively; second, the same qualitative temporal-block phenotype (persistence of early markers and failure to induce late markers) is produced by orthogonal manipu-lations that prolong G1/S or slow proliferation, including G1/S inhibitors and mitochondrial OxPhos inhibition (which increases G1/S occupancy), reported by others (*van den Ameele and Brand, 2019*). Thus, although we cannot rule out additional, gene-specific contributions of Cdk1 or Pav, the concor-dant effects of multiple, mechanistically distinct cell cycle perturbations make altered cell cycle dynamics the most parsimonious explanation. Definitive understanding of the molecular mechanism (e.g. whether mitosis-dependent chromatin remodeling, partitioning of factors during cytokinesis, or cell cycle-regulated signaling triggers late-factor induction) will require further experiments.

Our results suggest that only part of the temporal gene cascade depends on cell cycle progression. Depletion of Cdk1 disrupts the induction of late temporal factors (Syp, EcR) while leaving the onset of early switching factor Svp intact, indicating that the early expression of the temporal transition switching program can proceed independently of cell cycle progression. This distinction raises specific mechanistic possibilities: e.g., mitosis- or cytokinesis-coupled processes such as transcriptional reacti-vation, cell cycle-dependent post-translational modification of regulators, or daughter-cell-mediated signaling/partitioning could be required for late-factor induction but not for early *svp* activation. Defining the precise mechanism is beyond the scope of this study; however, our findings provide a testable framework for future experiments to determine how cell cycle-regulated changes in nuclear or cytoplasmic state enable the transition from early Imp/Chinmo to late Syp/EcR expression.

During cortical development, both cell cycle and extrinsic factors govern neuronal fate, and it was proposed that neuronal progenitors undergo cyclic changes in their ability to respond to extrinsic cues (*McConnell and Kaznowski, 1991*; *Okamoto et al., 2016*). Similarly, in *Drosophila*, Delta expression in the neighboring GMC and glia regulates Notch signaling, which is required for terminating neuro-genesis in the central brain (*Sood et al., 2024*). Further studies are needed to elucidate the mecha-nism by which the cell cycle influences EcR expression and to confirm the role of progeny within the lineage in regulating temporal gene expression transitions in NSCs.

# Materials and methods

**Key resources table**

| Reagent type (species) or resource | Designation | Source or reference | Identifiers | Additional information |
|---|---|---|---|---|
| Antibody | Anti-Deadpan (Rat, Monoclonal) | Abcam | Cat# ab195173; RRID:AB_2687586 | 1:300 |
| Antibody | Anti-GFP (Chicken, Polyclonal) | Aves Lab | Cat# GFP-1010; RRID:AB_2307313 | 1:1500 |
| Antibody | Anti-Asense (Rabbit, Clonality unknown) | Chen-Yu Lee | | 1:1000 |
| Antibody | Anti-IMP (Rat, Clonality unknown) | Claude Desplan | | 1:200 |
| Antibody | Anti β-GAL (Rabbit, Clonality unknown) | MP Biomedicals | Cat# 559762; RRID:AB_2335286 | 1:1000 |
| Antibody | Anti-EcR-B1 (Mouse, Monoclonal) | Carl Thummel | | 1:2000 |
| Antibody | Anti-Syncrip (Guinea pig, Polyclonal) | Ilan Davis | | 1:2000 |
| Chemical compound | 16% Paraformaldehyde | Electron Microscopy Sciences | Cat# 15710 | |
| Chemical compound | Triton X-100 | Sigma-Aldrich | Cat# T8787 | |
| Chemical compound | Apple Juice | Martinelli & Co | | |
| Chemical compound | Schneider's Insect medium | Sigma-Aldrich | Cat# S0146 | |
| Chemical compound | Agar | Sigma-Aldrich | Cat# A1296 | |
| Chemical compound | DPX mounting medium | Sigma-Aldrich | Cat# 06522 | |
| Chemical compound | Sucrose | Research Products International | Cat# 57-50-1 | |
| Chemical compound | Xylene | Fisher Scientific | Cat# 1330-20-7, 100-41-4 | |
| Genetic reagent (*D. melanogaster*) | hsFlp,UAS-mCD8GFP; Act-FRTstopFRT-GAL4 | Syed Lab | | |
| Genetic reagent (*D. melanogaster*) | UAS-pavRNAi | BDSC | BDSC_43963; RRID:BDSC_43963 | |
| Genetic reagent (*D. melanogaster*) | UAS-Cdk1RNAi | BDSC | BDSC_36117; RRID:BDSC_36117 | |
| Genetic reagent (*D. melanogaster*) | UAS-mCherryRNAi | BDSC | BDSC_35785; RRID:BDSC_35785 | |
| Genetic reagent (*D. melanogaster*) | Wor-GAL4, Ase-GAL80; 20xUASmcd8GFP | Syed Lab | | |
| Genetic reagent (*D. melanogaster*) | Wor-GAL4, Ase-GAL80; tub-GAL80ts | Syed Lab | | |
| Genetic reagent (*D. melanogaster*) | UAS-mcd8GFP (II) | BDSC | BDSC_5137; RRID:BDSC_5137 | |
| Genetic reagent (*D. melanogaster*) | 20xUAS-mcd8GFP (III) | BDSC | BDSC_32194; RRID:BDSC_32194 | |
| Genetic reagent (*D. melanogaster*) | svp-LacZ | BDSC | BDSC_26669; RRID:BDSC_26669 | |
| Software, algorthim | ImageJ | Fiji | RRID:SCR_002285 | Version: 2.9.0/1.53t |
| Software, algorthim | Adobe Illustrator | Adobe Systems | RRID:SCR_010279 | Version 24.005.20307 |

*Continued on next page*

*Continued*

| Reagent type (species) or resource | Designation | Source or reference | Identifiers | Additional information |
|---|---|---|---|---|
| Software, algorthim | GraphPad Prism 9 | GraphPad Software | RRID:SCR_002798; https://www.graphpad.com/ | |
| Software, algorthim | Imaris v9.9 and above | Oxford Instruments | RRID:SCR_007370; https://www.oxinst.com/news/imaris-launches-version-9.9-with-machine-learning-and-open-source-connections | |

## Fly genotype with associated figures

| Experimental line | Main | Supplementary |
|---|---|---|
| hsFlp,UAS-GFP;Act-FRTstopFRT-GAL4 crossed to UAS-mCherryRNAi | *Figure 2A, B, and C* | *Figure 2—figure supplement 1A-A' and E-E'* |
| hsFlp,UAS-GFP; Act-FRTstopFRT-GAL4 crossed to UAS-Cdk1RNAi | *Figure 2D, E, and F* | *Figure 2—figure supplement 1B-B' and F-F'* |
| hsFlp,UAS-GFP;act-FRTstopFRT-GAL4 crossed to UAS-pavRNAi | *Figure 2G, H, and I* | *Figure 2—figure supplement 1C-C' and G-G'* |
| Wor-GAL4, Ase-GAL80;tub-GAL80ts crossed to UAS-mcd8::GFP;UAS-mCherryRNAi | | *Figure 2—figure supplement 2A-A'''* |
| Wor-GAL4, Ase-GAL80;tub-GAL80ts crossed to UAS-mcd8::GFP;UAS-Cdk1RNAi | | *Figure 2—figure supplement 2B-B'''* |
| Wor-GAL4, Ase-GAL80;tub-GAL80ts crossed to UAS-mcd8::GFP;UAS-pavRNAi | | *Figure 2—figure supplement 2C-C'''* |
| Wor-GAL4,AseGAL80; 20xUASmcd8:GFP crossed to Svp-LacZ | *Figure 3A* | |
| WorGAL4,AseGAL80; 20xUASmcd8:GFP crossed to UAS-Cdk1-RNAi;Svp-LacZ | *Figure 3B* | |
| hsFlp,UASmCD8GFP; Act-FRTstopFRT-GAL4 crossed to UAS-mCherryRNAi | *Figure 4A, B, and C* | |
| hsFlp,UASmCD8GFP; Act-FRTstopFRT-GAL4 crossed to UAS-Cdk1RNAi | *Figure 4D, E, and F* | |
| hsFlp,UASGFP; Act-FRTstopFRT-GAL4 crossed to UAS-pavRNAi | *Figure 4G, H, and I* | |
| hsFlp,UASmCD8GFP; Act-FRTstopFRT-GAL4 crossed to UAS-mCherryRNAi | *Figure 5A, B, C, K, L, and M* | |
| hsFlp,UASmCD8 GFP; Act-FRTstopFRT-GAL4 crossed to UAS-Cdk1RNAi | *Figure 5D, E, F, N, O, and P* | |
| hsFlp,UASGFP; Act-FRTstopFRT-GAL4 crossed to UAS-pavRNAi | *Figure 5G, H, I, Q, R, and S* | |

## Immunohistochemistry

Larval brains were dissected in PBS and mounted on poly-D-lysine-coated coverslips. Samples were fixed for 23 min in 4% PFA in PBST and then washed in PBST for 3×20 min. Following this, the samples were blocked with 2% normal donkey serum and 2% normal goat serum (Jackson ImmunoResearch Laboratories, Inc) in PBST for 40 min at room temperature. Samples were then incubated in a primary antibody mix diluted in PBST overnight or for 1–2 days at 4°C. Primary antibodies were removed, and samples were thoroughly washed with PBST. Samples were then incubated in secondary antibodies for 2 hr at room temperature. Secondary antibodies were removed, and samples were washed in PBST. Samples were dehydrated with an ethanol series of 30%, 50%, 75%, and 100% ethanol, then incubated in xylene (Fisher Scientific X5-1) for 2×10 min. Samples were mounted onto slides with DPX (Sigma-Aldrich 06552) and cured for 3–4 days, then stored at 4°C until imaged.

## Image acquisition and analysis

Fluorescent images were acquired on both Zeiss LSM 710 and 800. Scale bars were given for all stacks within maximum intensity projection images. Brightness and contrast were adjusted in the

figure images individually for better visualization across the entire image in control and experimental samples. Statistics and bar graphs were computed using Prism 10.

### Figure preparation

Confocal images were prepared using Fiji or Imaris 10.0.0. Figures and schematics were assembled using Adobe Illustrator (2025).

### Generation of clones and quantification

To generate RNAi clones, *hsFLP;10xUASmcd8::GFP; ActFRTstopFRTGAL4* flies were crossed to the RNAi lines for the gene of interest or control *UAS-mCherryRNAi* line. Embryos were collected over a period of 4 hr. After hatching, larvae were collected and then heat-shocked in a 37°C water bath for 8 min and reared at 25°C until the desired time point. Temporal factor expression (Imp, Syp, EcR) was scored as positive or negative in individual neuroblasts. Because *pavRNAi* clones are frequently large and often multinucleated, reliable nuclear segmentation and per-nucleus intensity quantification were not feasible. To ensure reproducibility across samples, we normalized fluorescence signal intensity for each neuroblast to the mean intensity of neighboring wild-type neuroblasts imaged in the same field. A neuroblast was considered positive when its normalized intensity in the nucleus was at least 2× the local background. This scoring criterion was applied uniformly to all genotypes and time points. Only one clone was quantified per animal unless otherwise indicated. Experiments were replicated at least twice to ensure consistency.

## Acknowledgements

We thank Adil Wani, Asif Bakshi, Sen-Lin Lai, and Chris Doe for providing feedback on the manuscript. Thanks to Claude Desplan for generously sharing the antibodies. Stocks obtained from the Bloomington Drosophila Stock Center (NIH P40OD018537) were used in this study. We thank UNM CETI Biology Cell Biology Core for providing a confocal microscope facility. The research was supported by the Sloan Research Fellowship FG-2023-20617, 2023 BBRF Young Investigator Grant, McKnight Scholars Award, National Science Foundation CAREER Award IOS-2047020, Air Force Office of Scientific Research FA9550-24-1-0214, and NINDS R01NS136555 to MHS.

## Additional information

### Funding

| Funder | Grant reference number | Author |
|---|---|---|
| Alfred P. Sloan Foundation | FG-2023-20617 | Mubarak Hussain Syed |
| McKnight Foundation | McKnight Scholars Award | Mubarak Hussain Syed |
| Air Force Office of Scientific Research | FA9550-24-1-0214 | Mubarak Hussain Syed |
| National Science Foundation | IOS-2047020 | Mubarak Hussain Syed |
| National Institute of Neurological Disorders and Stroke | R01NS136555 | Mubarak Hussain Syed |
| Bloomington Drosophila Stock Center | NIH P40OD018537 | Mubarak Hussain Syed |

The funders had no role in study design, data collection and interpretation, or the decision to submit the work for publication.

### Author contributions

Gonzalo N Morales Chaya, Conceptualization, Formal analysis, Validation, Investigation, Visualization, Methodology, Writing – original draft; Mubarak Hussain Syed, Conceptualization, Supervision, Funding acquisition, Investigation, Project administration, Writing – review and editing

## Author ORCIDs
Gonzalo N Morales Chaya https://orcid.org/0009-0005-4263-1323
Mubarak Hussain Syed https://orcid.org/0000-0003-2424-175X

Reviewer #1 (Public review): https://doi.org/10.7554/eLife.108259.3.sa1
Reviewer #2 (Public review): https://doi.org/10.7554/eLife.108259.3.sa2
Reviewer #3 (Public review): https://doi.org/10.7554/eLife.108259.3.sa3
Author response https://doi.org/10.7554/eLife.108259.3.sa4

## Additional files

### Supplementary files
MDAR checklist

### Data availability
All relevant data and details of resources can be found within the article and its supplementary information.

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
