## [Editor Report · eLife Assessment]

This manuscript reports **important** findings indicating that cell cycle progression and cytokinesis both contribute to the transition from early to late neural stem cell fates. Loss-of-function experimental evidence **convincingly** shows that disrupting the cell cycle or cytokinesis can alter cell fate. This work sets the stage for future investigations into the underlying mechanisms linking the cell cycle to the expression of temporal factors controlling cell fate.

---

## [Referee Report · Reviewer #1 (Public review)]

Summary:

Drosophila larval type II neuroblasts generate diverse types of neurons by sequentially expressing different temporal identity genes during development. Previous studies have shown that transition from early temporal identity genes (such as Chinmo and Imp) to late temporal identity genes (such as Syp and Broad) depends on the activation of the expression of EcR by Seven-up (Svp) and progression through the G1/S transition of the cell cycle. In this study, Chaya and Syed examined if the expression of Syp and EcR is regulated by cell cycle and cytokinesis by knocking down CDK1 or Pav, respectively, throughout development or at specific developmental stages. They find that knocking down CDK1 or Pav either in all type II neuroblasts throughout the development or in single type neuroblast clones after larval hatching consistently leads to failure to activate late temporal identity genes Syp and EcR. To determine whether the failure of the activation of Syp and EcR is due to impaired Svp expression, they also examined Svp expression using a Svp-lacZ reporter line. They find that Svp is expressed normally in CDK1 RNAi neuroblasts. Further, knocking down CDK1 or Pav after Svp activation still leads to loss of Syp and EcR expression. Finally, they also extended their analysis to type I neuroblasts. They find that knocking down CDK1 or Pav, either at 0 hours or at 42 hours after larval hatching, also results in loss of Syp and EcR expression in type I neuroblasts. Based on these findings, the authors conclude that cycle and cytokinesis are required for the transition from early to late late temporal identity genes in both types of neuroblasts. These findings add mechanistic details to our understanding of the temporal patterning of Drosophila larval neuroblasts.

Strengths:

The data presented in the paper are solid and largely support their conclusion. Images are of high quality. The manuscript is well-written and clear.

Weaknesses:

The authors have addressed all the weaknesses in this revision.

---

## [Referee Report · Reviewer #2 (Public review)]

Summary:

Neural stem cells produce a wide variety of neurons during development. The regulatory mechanisms of neural diversity are based on the spatial and temporal patterning of neural stem cells. Although the molecular basis of spatial patterning is well-understood, the temporal patterning mechanism remains unclear. In this manuscript, the authors focused on the roles of cell cycle progression and cytokinesis in temporal patterning and found that both are involved in this process.

Strengths:

They conducted RNAi-mediated disruption on cell cycle progression and cytokinesis. As they expected, both disruptions affected temporal patterning in NSCs.

Weaknesses:

Although the authors showed clear results, they needed to provide additional data to support their conclusion sufficiently.

For example, they can examine the effects of cell cycle acceleration on the temporal patterning.

---

## [Referee Report · Reviewer #3 (Public review)]

Summary:

The manuscript by Chaya and Syed focuses on understanding the link between cell cycle and temporal patterning in central brain type II neural stem cells (NSCs). To investigate this, the authors perturb the progression of the cell cycle by delaying the entry into M phase and preventing cytokinesis. Their results convincingly show that temporal factor expression requires progression of the cell cycle in both Type 1 and Type 2 NSCs in the Drosophila central brain. Overall, this study establishes an important link between the two timing mechanisms of neurogenesis.

Strengths:

The authors provide solid experimental evidence for the coupling of cell cycle and temporal factor progression in Type 2 NSCs. The quantified phenotype shows an all-or-none effect of cell cycle block on the emergence of subsequent temporal factors in the NSCs, strongly suggesting that both nuclear division and cytokinesis are required for temporal progression. The authors also extend this phenotype to Type 1 NSCs in the central brain, providing a generalizable characterization of the relationship between cell cycle and temporal patterning.

Weaknesses:

One major weakness of the study is that the authors do not explore the mechanistic relationship between cell cycle and temporal factor expression. Although their results are quite convincing, they do not provide an explanation as to why Cdk1 depletion affects Syp and EcR expression but not the onset of svp. This result suggests that at least a part of the temporal cascade in NSCs is cell-cycle independent which isn't addressed or sufficiently discussed.

---

## [Author Response]

The following is the authors’ response to the original reviews.

**Public Reviews:**

**Reviewer #1 (Public review):**
Summary:Drosophila larval type II neuroblasts generate diverse types of neurons by sequentially expressing different temporal identity genes during development. Previous studies have shown that the transition from early temporal identity genes (such as Chinmo and Imp) to late temporal identity genes (such as Syp and Broad) depends on the activation of the expression of EcR by Seven-up (Svp) and progression through the G1/S transition of the cell cycle. In this study, Chaya and Syed examined whether the expression of Syp and EcR is regulated by cell cycle and cytokinesis by knocking down CDK1 or Pav, respectively, throughout development or at specific developmental stages. They find that knocking down CDK1 or Pav either in all type II neuroblasts throughout development or in single-type neuroblast clones after larval hatching consistently leads to failure to activate late temporal identity genes Syp and EcR. To determine whether the failure of the activation of Syp and EcR is due to impaired Svp expression, they also examined Svp expression using a Svp-lacZ reporter line. They find that Svp is expressed normally in CDK1 RNAi neuroblasts. Further, knocking down CDK1 or Pav after Svp activation still leads to loss of Syp and EcR expression. Finally, they also extended their analysis to type I neuroblasts. They find that knocking down CDK1 or Pav, either at 0 hours or at 42 hours after larval hatching, also results in loss of Syp and EcR expression in type I neuroblasts. Based on these findings, the authors conclude that cycle and cytokinesis are required for the transition from early to late temporal identity genes in both types of neuroblasts. These findings add mechanistic details to our understanding of the temporal patterning of Drosophila larval neuroblasts.Strengths:The data presented in the paper are solid and largely support their conclusion. Images are of high quality. The manuscript is well-written and clear.

We appreciate the reviewer’s detailed summary and recognition of the study’s strengths.

Weaknesses:The quantifications of the expression of temporal identity genes and the interpretation of some of the data could be more rigorous.(1) Expression of temporal identity genes may not be just positive or negative. Therefore, it would be more rigorous to quantify the expression of Imp, Syp, and EcR based on the staining intensity rather than simply counting the number of neuroblasts that are positive for these genes, which can be very subjective. Or the authors should define clearly what qualifies as "positive" (e.g., a staining intensity at least 2x background).

We thank the reviewer for this helpful suggestion. In the new version, we have now clarified how positive expression was defined and added more details of our quantification strategy to the Methods section (page 11, lines 380-388; lines 426-434 in tracked changes file). Fluorescence intensity for each neuroblast was normalized to the mean intensity of neighboring wild-type neuroblasts imaged in the same field. A neuroblast was considered positive for a given factor when its normalized nuclear intensity was at least 2× the local background. This scoring criterion was applied uniformly across all genotypes and time points. All quantifications were performed on the raw LSM files in Fiji prior to assembling the figure panels.

(2) The finding that inhibiting cytokinesis without affecting nuclear divisions by knocking down Pav leads to the loss of expression of Syp and EcR does not support their conclusion that nuclear division is also essential for the early-late gene expression switch in type II NSCs (at the bottom of the left column on page 5). No experiments were done to specifically block the nuclear division in this study specifically. This conclusion should be revised.

We blocked both cell cycle progression and cytokinesis, and both these manipulations affected temporal gene transitions, suggesting that both cell cycle and cytokinesis are essential. To our knowledge, no mechanism/tool exists that selectively blocks nuclear division while leaving cell cycle progression intact. We have added more clarification on page 4, line 123 onwards (lines 126 onwards in tracked changes file).

(3) Knocking down CDK1 in single random neuroblast clones does not make the CDK1 knockdown neuroblast develop in the same environment (except still in the same brain) as wild-type neuroblast lineages. It does not help address the concern whether "type 2 NSCS with cell cycle arrest failed to undergo normal temporal progression is indirectly due to a lack of feedback signaling from their progeny", as discussed (from the bottom of the right column on page 9 to the top of the left column on page 10). The CDK1 knockdown neuroblasts do not divide to produce progeny and thus do not receive a feedback signal from their progeny as wild-type neuroblasts do. Therefore, it cannot be ruled out that the loss of Syp and EcR expression in CDK1 knockdown neuroblasts is due to the lack of the feedback signal from their progeny. This part of the discussion needs to be clarification.

Thanks to the reviewer for raising this critical point. We agree and have added more clarification of our interpretations and limitations to our studies in the revised text on page 8, line 278-282 (lines 296-300 in tracked changes file)

(4) In Figure 2I, there is a clear EcR staining signal in the clone, which contradicts the quantification data in Figure 2J that EcR is absent in Pav RNAi neuroblasts. The authors should verify that the image and quantification data are consistent and correct.

When cytokinesis is blocked using pav-RNAi, the neuroblasts become extremely large and multinucleated. In some large pav RNAi clones, we observed a weak EcR signal near the cell membrane. However, more importantly, none of the nuclear compartments showed detectable EcR staining, where EcR is typically localized. We selected a representative nuclear image for the figure panel. To clarify this observation, we have now added an explanatory note to the discussion section on page 8, lines 283-291 (lines 301-309 in tracked changes file).

**Reviewer #2 (Public review):**
Summary:Neural stem cells produce a wide variety of neurons during development. The regulatory mechanisms of neural diversity are based on the spatial and temporal patterning of neural stem cells. Although the molecular basis of spatial patterning is well-understood, the temporal patterning mechanism remains unclear. In this manuscript, the authors focused on the roles of cell cycle progression and cytokinesis in temporal patterning and found that both are involved in this process.Strengths:They conducted RNAi-mediated disruption on cell cycle progression and cytokinesis. As they expected, both disruptions affected temporal patterning in NSCs.

We appreciate the reviewer’s positive assessment of our experimental results.

Weaknesses:Although the authors showed clear results, they needed to provide additional data to support their conclusion sufficiently.For example, they need to identify type II NSCs using molecular markers (Ase/Dpn).The authors are encouraged to provide a more detailed explanation of each experiment. The current version of the manuscript is difficult for non-expert readers to understand.

Thanks for your feedback. We have now included a detailed description of how we identify type II NSCs in both wild-type and mutant clones. We have also added a representative Asense staining to clearly distinguish type 1 (Ase^+^) from type 2 (Ase^-^) NSCs see Figure S1. We have also added a resources table explaining the genotypes associated with each figure, which was omitted due to an error in the previous version of the manuscript.

**Reviewer #3 (Public review):**
Summary:The manuscript by Chaya and Syed focuses on understanding the link between cell cycle and temporal patterning in central brain type II neural stem cells (NSCs). To investigate this, the authors perturb the progression of the cell cycle by delaying the entry into M phase and preventing cytokinesis. Their results convincingly show that temporal factor expression requires progression of the cell cycle in both Type 1 and Type 2 NSCs in the Drosophila central brain. Overall, this study establishes an important link between the two timing mechanisms of neurogenesis.Strengths:The authors provide solid experimental evidence for the coupling of cell cycle and temporal factor progression in Type 2 NSCs. The quantified phenotype shows an all-ornone effect of cell cycle block on the emergence of subsequent temporal factors in the NSCs, strongly suggesting that both nuclear division and cytokinesis are required for temporal progression. The authors also extend this phenotype to Type 1 NSCs in the central brain, providing a generalizable characterization of the relationship between cell cycle and temporal patterning.

We thank the reviewer for recognizing the robustness of our data linking the cell cycle to temporal progression.

Weaknesses:One major weakness of the study is that the authors do not explore the mechanistic relationship between the cell cycle and temporal factor expression. Although their results are quite convincing, they do not provide an explanation as to why Cdk1 depletion affects Syp and EcR expression but not the onset of svp. This result suggests that at least a part of the temporal cascade in NSCs is cell-cycle independent, which isn't addressed or sufficiently discussed.

Thank you for bringing up this important point. We are equally interested in uncovering the mechanism by which the cell cycle regulates temporal gene transitions; however, such mechanistic exploration is beyond the scope of the present study. Interestingly, while the temporal switching factor Svp is expressed independently of the cell cycle, the subsequent temporal transitions are not. We have expanded our discussion on this intriguing finding (page 9, line 307-315; lines 345-355 in tracked changes file). Specifically, we propose that svp activation marks a cell-cycle–independent phase, whereas EcR/Syp induction likely depends on cell-cycle–coupled mechanisms, such as mitosis-dependent chromatin remodeling or daughter-cell feedback. Although further dissection of this mechanism lies beyond the current study, our findings establish a foundation for future work aimed at identifying how developmental timekeeping is molecularly coupled to cell-cycle progression.

**Recommendations for the authors:**

**Reviewer #1 (Recommendations for the authors):**
(1) Figure 1 C and D, it would be better to put a question mark to indicate that these are hypotheses to be tested.

We appreciate this suggestion and have added question marks in Figure 1C and 1D to clearly indicate that these panels represent hypotheses under investigation clearly.

(2) Figure 2A-I, Figure 4A-I, Figure 5A-I and K-S, in addition to enlarged views of single type II neuroblasts, it would be more convincing to include zoomed-out images of the entire larval brain or at least a portion of the brain to include neighboring wild-type type II neuroblasts as internal controls. Also, it would be ideal to show EcR staining from the same neuroblasts as IMP and Syp staining.

We thank the reviewer for this valuable input. In our imaging setup, the number of available antibody channels was limited to four (anti-Ase, anti-GFP, anti-Syp, and antiImp). Adding EcR in the same sample was therefore not technically possible, we performed EcR staining separately.

(3) The authors cited "Syed et al., 2024" (in the middle of the right column on page 5), but this reference is missing in the "References" section and should be added.

The missing citation has been added to the reference section.

(4) It would be better to include Ase staining in the relevant figure to indicate neuroblast identity as type I or type II.

We agree and now include representative Ase staining for both type 1 and type 2 NSC clones in Figure S1, along with corresponding text updates that describe these markers.

**Reviewer #2 (Recommendations for the authors):**
Major comments(1) The present conclusion relies on the results using Cdk1 RNAi and pav RNAi. It is still possible that Cdk1 and Pav are involved in the regulation of temporal patterning independent of the regulation of cell cycle or cytokinesis, respectively. To avoid this possibility, the authors need to inhibit cell cycle progression or cytokinesis in another alternative manner.

We thank the reviewer for raising this important point. While we cannot completely exclude gene-specific, cell-cycle-independent roles for Cdk1 or Pav, we observe consistent phenotypes across several independent manipulations that slow or block the cell cycle. Also, earlier studies using orthogonal approaches that delay G1/S (Dacapo/Rbf) or impair mitochondrial OxPhos (which lengthens G1/S; van den Ameele & Brand, 2019) produce similar temporal delays. These concordant phenotypes strongly support the interpretation that altered cell-cycle progression—rather than specific roles of a single gene—is the primary cause of the defect. While we cannot exclude additional, gene-specific effects of Cdk1 or Pav, the concordant phenotypes across independent perturbations make the cell-cycle disruption model the most parsimonious interpretation. We have clarified this reasoning in the discussion section on pages 8-9, lines 293-305 (lines 311-343 in tracked changes file).

(2) To reach the present conclusion, the authors need to address the effects of acceleration of cell cycle progression or cytokinesis on temporal patterning.

We thank the reviewer for this insightful suggestion. To our knowledge, there are currently no established genetic tools that can specifically accelerate cell-cycle progression in Drosophila neuroblasts. However, our results demonstrate that blocking the cell cycle impairs the transition from early to late temporal gene expression. These findings suggest that proper cell-cycle progression is essential for the transition from early to late temporal identity in neuroblasts.

Minor comments(3) P3L2 (right), ... we blocked the NSC cell cycle...How did they do it?Which fly lines were used?Why did they use the line?

These details are now included in the Materials and Methods and the Resource Table (pages 11-13). We used Wor-Gal4, Ase-Gal80 to drive UAS-Cdk1RNAi and UASpavRNAi in type 2 NSCs

(4) P5L1(left), ... we used the flip-out approach...Why did they conduct it?Probably, the authors have reasons other than "to further ensure."

We have clarified in the text on page 4, lines 137-139, that the flip-out approach was used to generate random single-cell clones, enabling quantitative analysis of type 2 NSCs within an otherwise wild-type brain.

(5) P5L8(left), ... type 2 hits were confirmed by lack of the type 1 Asense... The authors must examine Deadpan (Dpn) expression as well. Because there are a lot of Asense (Ase) negative cells in the brain (neurons, glial cell, and neuroepithelial cells).Type II NSCs can be identified as Dpn+/Ase- cells.

We agree that Dpn is a helpful marker. However, we reliably distinguished type II NSCs by their lack of Ase and larger cell size relative to surrounding neurons and glia, which are smaller in size and located deeper within the clone. These differences, together with established lineage patterns, allow unambiguous identification of type 2 NSCs across all genotypes. We have now added representative type I and type 2 NSC clones to the supplemental figure S1 (E-G’) with Asense stains to demonstrate how we differentiate type I from type II NSCs.

(6) P5L32(left), To do this, we induced...This sentence should be made more concise.Please rephrase it.

The sentence has been rewritten for clarity and concision.

(7) P5L42(left), ...lack of EcR/Syp expression (Figure 2). However, EcR expression is still present (Figure 2I).

In some large pavRNAi clones, a weak EcR signal can be observed near the cell membrane; however, none of the nuclear compartments—where EcR is typically localized—show detectable staining. We selected a representative nuclear image for the figure and addressed this observation on page 8, lines 283-291 (lines 301-309 in tracked changes file).

(8) P7L29(left), ......had persistent Imp expression...Imp expression is faint compared to that in Figure 2G.The differences between Figures 2G and 3G should be discussed.

We thank the reviewer for this comment. We have added a note in the Methods section clarifying that brightness and contrast were adjusted per panel for optimal visualization; thus, apparent differences in signal intensity do not reflect biological variation. Fluorescence intensity for each neuroblast was normalized to the mean intensity of neighboring wild-type neuroblasts imaged in the same field. A neuroblast was considered Imp-positive when its normalized nuclear intensity was at least 2× the local background. This scoring criterion was applied uniformly across all genotypes and time points. All quantifications were performed on the raw LSM files in Fiji prior to assembling the figure panels.

(9) P8 (Figure 5)The Imp expression is faint compared to that in Figure 5Q.The difference between Figure 5G and 5Q should be discussed further.

As mentioned above, we have clarified our image processing approach in the Methods section to explain any differences in signal appearance between these figures.

(10) P10 Materials and MethodsThe authors did not mention the fly lines used. This is very important for the readers.

We thank the reviewer for bringing this oversight to our attention. The Resource Table was inadvertently omitted from the initial submission. The complete list of fly lines and reagents used in this study is now provided in the updated Resource Table.

**Reviewer #3 (Recommendations for the authors):**
Major points(1) The authors mention that the heat-shock induction at 42ALH is well after svp temporal window and therefore the cell cycle block independently affects Syp and EcR expression. However, Figure 3 shows svp-LacZ expression at 48ALH. If svp expression is indeed transient in Type 2 NSCs, then this must be validated using an immunostaining of the svp-LacZ line with svp antibody. This is crucial as the authors claim that cell cycle block doesn't affect does affect svp expression and is required independently.

We thank the reviewer for bringing this important issue to our attention. As noted, Svp protein is expressed transiently and stochastically in type 2 NSCs (Syed et al., 2017), making direct antibody quantification challenging upon cell cycle block. Consistent with previous work (Syed et al., 2017), we used the svp-LacZ reporter line to visualize stabilized Svp expression, which reliably captures Svp expression in type 2 NSCs (Syed et al., 2017 https://doi.org/10.7554/eLife.26287, and Dhilon et al., 2024 https://doi.org/10.1242/dev.202504).

(2) The authors have successfully slowed down the cell cycle and showed that it affects temporal progression. However, a converse experiment where the cell cycle is sped up in NSCs would be an important test for the direct coupling of temporal factor expression and cell cycle, wherein the expectation would be the precocious expression of late temporal factors in faster cycle NSCs.

We agree that such an experiment would be ideal. However, as noted above (Reviewer #2 comment 2), to our knowledge, no suitable tools currently exist to accelerate neuroblast cell-cycle progression without pleiotropic effects.

Minor pointThe authors must include Ray and Li (https://doi.org/10.7554/eLife.75879) in the references when describing that "...cell cycle has been shown to influence temporal patterning in some systems,...".

We thank the reviewer for this helpful suggestion. The cited reference (Ray and Li, eLife, 2022) has now been included and appropriately referenced in the revised manuscript.